# Histological Scores Validate the Accuracy of Hepatic Iron Load Measured by Signal Intensity Ratio and R2* Relaxometry MRI in Dialysis Patients

**DOI:** 10.3390/jcm9010017

**Published:** 2019-12-19

**Authors:** Guy Rostoker, Mireille Laroudie, Raphaël Blanc, Mireille Griuncelli, Christelle Loridon, Fanny Lepeytre, Clémentine Rabaté, Yves Cohen

**Affiliations:** 1Division of Nephrology and Dialysis (Service Néphrologie et de Dialyse), Ramsay-Santé, Hôpital Privé Claude Galien, 91480 Quincy-Sous-Sénart, France; 2Collège de Médecine des Hôpitaux de Paris, 75005 Paris, France; 3Histopathology Laboratory ACP Bièvres (Laboratoire d’Anatomie et de Cytologie Pathologiques (ACP) Bièvres), Les Ulis—Praxea Diagnostics, 91300 Massy, France; 4Division of Angiography (Service de Radiologie Interventionnelle), Ramsay-Santé, Hôpital Privé Claude Galien, 91480 Quincy-Sous-Sénart, France; 5Division of Radiology (Service de Radiologie), Ramsay-Santé, Hôpital Privé Claude Galien, 91480 Quincy-Sous-Sénart, France

**Keywords:** MRI, signal intensity ratio, R2* relaxometry, Perls’ stain, Scheuer’s classification, Deugnier’s and Turlin’s histological scoring, dialysis

## Abstract

Almost all haemodialysis patients are treated with parenteral iron to compensate for blood loss and to allow the full therapeutic effect of erythropoiesis-stimulating agents. Iron overload is an increasingly recognised clinical situation diagnosed by quantitative magnetic resonance imaging (MRI). MRI methods have not been fully validated in dialysis patients. We compared Deugnier’s and Turlin’s histological scoring of iron overload and Scheuer’s classification (with Perls’ stain) with three quantitative MRI methods for measuring liver iron concentration (LIC)—signal intensity ratio (SIR), R2* relaxometry, and R2* multi-peak spectral modelling (Iterative Decomposition of water and fat with Echo Asymmetry and Least-squares estimation (IDEAL-IQ^®^)) relaxometry—in 16 haemodialysis patients in whom a liver biopsy was formally indicated for medical follow-up. LIC MRI with these three different methods was highly correlated with Deugnier’s and Turlin’s histological scoring (SIR: *r* = 0.8329, *p* = 0.0002; R2* relaxometry: *r* = −0.9099, *p* < 0.0001; R2* relaxometry (IDEAL-IQ^®^): *r* = −0.872, *p* = 0.0018). Scheuer’s classification was also significantly correlated with these three MRI techniques. The positive likelihood ratio for the diagnosis of abnormal LIC by Deugnier’s histological scoring was > 62 for the three MRI methods. This study supports the accuracy of quantitative MRI methods for the non-invasive diagnosis and follow-up of iron overload in haemodialysis patients.

## 1. Introduction 

The discovery of epoetin 30 years ago represented a therapeutic revolution in end-stage kidney disease (ESKD), allowing the partial correction of anaemia in most patients and the avoidance of blood transfusion with its corollary of human leucocyte antigen (HLA) sensitisation and transfusion-related iron overload, deeply transforming the quality of life and clinical outcomes of ESKD patients [1,2]. Almost all haemodialysis patients receive parenteral iron to ensure efficient erythropoiesis with erythropoiesis-stimulating agents (ESAs) in order to overcome functional iron deficiency and to compensate true iron deficiency related to important blood loss in the setting of dialysis [1,2].

Intravenous (IV) iron therapy, together with ESAs, forms the backbone of anaemia treatment in ESKD due to its convenience (infusion during dialysis sessions), superiority over oral preparations (poorly tolerated) and cost savings of approximately 20% on expensive ESA molecules [1,3,4].

Iron overload in ESKD was considered to be a classical complication of iterative blood transfusions in the pre-ESA era and was believed until recently to be exceptional among haemodialysis patients in the current ESA era. However, it is now an increasingly recognised clinical problem affecting up to 66% of haemodialysis patients [3,4,5,6,7,8,9,10,11]. 

Since the liver is the main iron storage site in humans, and because liver iron concentration (LIC) correlates closely with total body iron stores in patients with genetic haemochromatosis and secondary haemosiderosis, hepatic magnetic resonance imaging (MRI) has become the gold standard method for estimating and monitoring iron stores in patients with iron-related disorders [12,13,14,15].

There are currently four validated MRI methods for determining LIC: signal intensity ratio (SIR) MRI (according to Rennes University), R2 relaxometry, R2* relaxometry (classical), and R2* relaxometry with multi-peak spectral modelling. These four MRI methods have been validated exclusively in cohorts of non-renal patients (with secondary haemosiderosis, genetic haemochromatosis or other liver diseases) who underwent liver biopsy for biochemical iron assays [15,16,17,18,19].

Studies using MRI to estimate LIC in haemodialysis patients have recently shown a strong link between infused iron dose and risk of iron overload and have revealed an epidemic prevalence of haemodialysis-associated haemosiderosis [3,7,8,9,10,11].

The recent Proactive IV Iron Therapy in Haemodialysis Patients (PIVOTAL) trial has demonstrated strong evidence of a benefit of IV iron in the treatment of anaemia in ESKD, with decreased hospitalisation rates related to cardiac insufficiency and reduced ESA cost [20]. Conversely, while the PIVOTAL study will result in an increased use of parenteral iron in clinical practice, there is a need to take into account the double-edged sword of iron therapy [21], especially hepatic iron accumulation, which has been associated with increased hepcidin production and a risk of destabilising atheromatous plaques, triggering cardiovascular events [9,10,11,22] and inducing or worsening fatty liver disease [23], together with the increased mortality shown by Dialysis Outcomes and Practice Patterns Study (DOPPS) in ESKD patients receiving over 300 mg monthly IV iron [24].

The Kidney Disease Improvement of Global Outcomes’ (KDIGO) controversies conference on iron management in chronic kidney disease, held in San Francisco in March 2014, called for an agenda of research on the topic of iron overload, especially with MRI techniques [25]. Validation of LIC determination using MRI methods in dialysis patients, by comparison with liver biopsy, appears to be of high scientific importance but cannot be solved by a prospective trial due to ethical concerns related to the invasiveness and risks of liver biopsy in the frail ESKD population.

In this study, we compared two validated histological scoring of liver iron overload [26,27] with three quantitative MRI methods for measuring liver iron content in 16 haemodialysis patients in whom liver biopsy was formally indicated for their medical follow-up [16,18,19].

## 2. Materials and Methods

### 2.1. Patients and Dialysis

This prospective, cross-sectional and longitudinal study was started on 31 January 2005. The aim was to study iron stores by MRI and its determinants. The study received technical and ethical approval from the Drug, Devices and Clinical Trials Committee of our institution (COMEDIMS Claude Galien, 9 December 2004 and 15 February 2013) and was conducted in accordance with the declaration of Helsinki.

Dialysis patients were enrolled if they were free from overt inflammation or malnutrition and were undergoing either chronic intermittent haemodialysis or haemodiafiltration three times a week or peritoneal dialysis at the dialysis unit of Claude Galien hospital. All patients gave their written informed consent. The patients were treated for anaemia according to European best practice guidelines [28]. The inclusion and exclusion criteria of the study and the treatment of anaemia have been described previously [10].

We calculated the duration of exposition to IV iron therapy (in months) as its magnitude (cumulative IV iron dose (in gram) and average monthly dose (in milligram) [10].

The study is registered under International Standard Randomised Controlled Trial Number (ISRCTN): 80100088 [10]. The study was also declared to the French Institute of Health Data (INDS: Institut National des Données de Santé) as an observational study (category MR-4 of Jardé law) and its database was declared to the devoted French commission (CNIL) under the successive numbers 1875675 (until 2018) and 2214279 (since 2019).

### 2.2. Quantification of Hepatic Iron Stores by MRI

MRI measurements were all made by the same senior radiologist (Y.C.), who was unaware of the patients’ medical history and iron biomarker values for follow-up of iron therapy, on an Optima™ MR450w MRI machine (GE Medical Systems, Milwaukee, WI, USA) operating at a field strength of 1.5 Tesla. Wherever possible, patients on iron therapy received their last iron dose at least one week before MRI. During the same session, LIC was assessed by three quantitative MRI methods simultaneously—SIR according to Rennes University, R2* relaxometry (classical), and R2* relaxometry with multi-peak spectral modelling (IDEAL-IQ^®^)—according to the manufacturer (for the latter two methods—General Electric medical systems) [16,18,19]. Furthermore, an analysis of liver for dysmorphia was performed by Y.C. using the T1 sequence (with homemade software) before any iron measurement. Areas of cyst-free liver were examined in patients with genetic polycystosis and the less dismorphic areas were chosen in patients with liver disease.

SIR MRI: The method used for measurement of LIC was based on T1 and T2* contrast imaging (without gadolinium), as established by Gandon et al. at Rennes University [16]. We acquired five weighted gradient-recalled echo sequences of the liver (GRE T1, PD, T2, T2+ and T2++) with a repetition time of 120 ms, bandwidth of 12.5 kHz, matrix of 256 × 128, field of view of 40 cm and slice thickness of 10 mm. Measurements were made in five regions of interest (ROIs) larger than 1 cm^2^ (usually three on the right liver and two on paraspinal muscles on the same slice), and we calculated the liver-to-muscle ratio. We used the software algorithm freely provided by Rennes University (http://www.radio.univ-rennes1.fr) to determine hepatic iron content. LIC is expressed in µmol/g of dry liver. Normal LIC values were set at < 40 µmol/g according to Rennes University [15,16].

R2* relaxometry (classical): Liver R2* was measured according Wood et al. (2005) [18] using a single-gradient echo sequence on a sole mid-hepatic slice; echo time (TE) was automatically stepped at 1.1 ms intervals from 0.9 to 8.8 ms in a single breath-hold. The other imaging parameters were as follows: a field of view of 42 × 34, flip angle of 25°, repetition time (TR) of 35 ms, matrix of 128 × 128, slice thickness of 8 mm and a bandwidth of 125 kHz. The images were processed by using the software provided by the manufacturer (GE Healthcare) to generate T2*/R2* maps. Measurements were made in two ROIs (of at least 1 cm^2^) frequently on the right liver. The results are expressed as R2* in Hertz (Hz) and converted in T2* (ms) (T2* in ms = 1000/R2* in Hz) with a local homemade calculator [15,18]. Normal T2* LIC values were set at ≥ 15 ms according to Paisant et al. [15].

R2* relaxometry (IDEAL-IQ^®^) (multi-peak spectral modelling): This technique has been performed in our radiology department since May 2015. The IDEAL-IQ^®^ algorithm is Food and Drug Administration (FDA)-validated software of GE Healthcare, which uses multi-peak spectral modelling, allowing simultaneous reliable liver fat and iron quantification [19,29]. The IDEAL-IQ imaging technique is a triglyceride fat and water separation technique that acquires multiple images of the anatomy at separate echo times to calculate the phase differences and determine triglyceride fat and water content per pixel.

The results were expressed as R2* in Hz and converted to T2* in ms (T2* in ms = 1000/R2* in Hz) and normal LIC-T2*-IDEAL-IQ^®^ values were also set by analogy with relaxometry (classical) T2* ≥ 15 ms.

The parameters of the IDEAL-IQ^®^ sequence were as follows: TR (repetition time) of 14 ms; TE of 6; number of echoes of 6, ranging from 1.1 to 6.38 ms; field of view of 35–40 cm; matrix size of 224 × 192; pixel bandwidth of 100 Hz; flip angle of 8; slice thickness of 10 mm; and space between slices of 5 mm. The sequence was acquired in a single breath-hold time of approximately 25 s in all patients. The images were processed by using the software provided by the manufacturer (IDEAL-IQ^®^; GE Healthcare) to generate both fat fraction maps and T2*/R2* maps. At first, the radiologist chooses the best image of liver and determines two ROIs on the organ—the positioning of the measurements being frequently on the right liver. The size of the ROIs is variable but must be at least 1 cm^2^.

Splenic iron content: Splenic iron load was also assessed during the same session by R2* relaxometry and expressed as R2* in Hz and converted to T2* in ms (T2* in ms = 1000/R2* in Hz) (normal value of spleen is T2* ≥ 15 ms according to Ghoti et al.) [9].

### 2.3. Quantitative Liver Histology

Liver specimens obtained by transjugular biopsy (*n* = 16), wedge biopsy (*n* = 1) or partial hepatectomy (*n* = 1) were fixed in neutral-buffered formalin, zinc chloride Zenker’s or Hollande’s fixative and embedded in paraffin. Haematoxylin–eosin, Masson’s trichrome and Perls’ stain were available for each specimen [26]. Each biopsy was examined for the presence of specific pathological features [26].

Hepatic iron was assessed quantitatively (by M.L., a senior pathologist who had no knowledge of the patients’ medical history and liver MRI results) by light microscopy on slides stained with Perls’ stain according to Deugnier’s and Turlin’s histological scoring (total iron score: 0–60) after determining hepatocytic iron score (0–36), sinusoidal iron score (0–12) and portal iron score (0–12); an abnormal score of stainable liver iron is equal to ≥ 6 (normal iron load is 0–5 at total iron score) (Table 1) [26]. Deugnier’s and Turlin’s histological scoring has been validated in both genetic haemochromatosis and secondary haemosiderosis by comparison with biochemical quantification of liver iron [30,31,32,33,34] and is considered the gold standard for histological quantification in iron-related disorders [26]. This method has been proposed to differentiate between heterozygotes and homozygotes in genetic haemochromatosis when biochemical quantification of liver iron was not available in the pre-genetic testing era [33]. Finally, Deugnier’s and Turlin’s histological scoring has also been validated for the evaluation of mild iron load by comparison with biochemical quantification of liver iron in non-haemochromatotic liver disease [34].

Deugnier’s and Turlin’s histological scoring is the finalised outcome of the previous histological classification of Brissot et al. at Rennes University, which was shown in the 1980s to correlate closely with iron liver measured chemically in iron overloaded dialysis patients during the pre-ESA era [35,36].

Hepatic iron was also graded (by M.L.) on a semi-quantitative scale of 0–4 according to Scheuer’s classification, which solely quantifies hepatocyte iron [27]. The normal grade of stainable liver iron is 0–1, while grade 4 is the degree seen in fully developed untreated haemochromatosis (Table 2) [26,27].

### 2.4. Statistical Analyses

Correlations between absolute LIC values by the three MRI methods and scores by the two histological classifications were analysed with Spearman’s rank correlation test [37]. Prism 8 software (GraphPad, San Diego, CA, USA) was used for all tests and *p*-values < 0.05 were considered to denote statistical significance [37].

We also assessed the accuracy of the three MRI methods and Deugnier’s and Turlin’s histological scoring for the diagnosis of iron overload in haemodialysis patients by calculating sensitivity, specificity, positive predictive value (PPV), negative predictive value (NPV), the positive likelihood ratio (PLR) and posterior probability [38,39]. The threshold of normality was set at < 6 for Deugnier’s and Turlin’s histological scoring [33,34]; for MRI methods, normal LIC was set at < 40 µmol/g for SIR [16] and ≥ 15 ms for R2* relaxometry (classical) and R2* relaxometry (IDEAL-IQ^®^) [15]. The prevalence of iron overload in haemodialysis patients was considered to be 66% (99% CI: 0.60–0.71), based on a pooled analysis of 11 radiological studies (one with susceptometry and 10 with quantitative MRI) performed worldwide in 500 haemodialysis patients [11]. Sensitivity and specificity were calculated on XLSTAT (Addinsoft, Paris, France), whereas the PPV, NPV, PLR and posterior probability were determined on the Bayesian calculator QuickCalcs (GraphPad, San Diego, CA, USA). A PLR of > 10 was considered to provide strong evidence for the diagnosis of iron overload in haemodialysis patients in most circumstances [39]. Fagan’s nomogram was established for each MRI method using the Diagnostic Test Calculator from Alan Schwartz [40,41].

## 3. Results

### 3.1. Characteristics of the Study Population

Sixteen patients were recruited (eight women and eight men, with a median age of 58 years (range: 36–74)) between 27 March 2013 and 9 February 2018. The median haemodialysis duration was 18 months (range: 3–140). The original nephropathy was as follows: diabetes mellitus (*n* = 6), hypertensive nephropathy (*n* = 3), chronic pyelonephritis (*n* = 1), congenital uropathy (*n* = 3), genetic segmental sclerosis (*n* = 1), systemic lupus erythematosus (*n* = 1) and ANCA vasculitis (*n* = 1). Their median duration of exposition to iron therapy until MRI and liver biopsy was 17.5 months (range: 4–140) and their median cumulative iron dose was 2.7 g (range: 0.18–13.4), whereas their median average monthly iron dose was 191.2 mg (range: 30.8–303.8) in 12 out of these 16 patients in whom an extensive medical chart analysis could be performed.

Fourteen patients underwent a transjugular biopsy with a Menghini needle (all performed by R.B.); of note, two of these patients later required a second transjugular hepatic biopsy (performed by R.B.) (and MRI) for the follow-up of their liver disease. No complications occurred during the procedure including haemorrhagic events. One patient had a wedge biopsy, and another underwent a partial surgical hepatectomy. Thus, the total number of liver biopsies analysed was 18 in these 16 patients. For the statistical analysis, each biopsy was analysed separately. Deugnier’s and Turlin’s histological scoring was performed successfully on 15 liver biopsies but failed in three biopsies due to the small size of the histological specimen obtained by transjugular biopsy. Scheuer’s classification was performed successfully in all 18 liver biopsies. The median Deugnier score was 6 (range: 0–41) and 10/15 biopsies had increased LIC at histology (Deugnier score > 5). The median Scheuer grade was 2 (range: 0–3) and 13/18 biopsies had increased LIC at histology with this second histological scoring system (Scheuer score > 1).

The final liver histological diagnoses were as follows: hepatic haemosiderosis, *n* = 7 (with a mixed pattern, *n* = 3; with a reticuloendothelial pattern, *n* = 3; with hepatocytic (parenchymal) pattern, *n* = 1); large cell hepatic dysplasia associated with reticuloendothelial haemosiderosis, *n* = 1; alcoholic fibrosteatosis, *n* = 2; alcoholic cirrhosis, *n* = 1; steatosis, *n* = 1; mycophenolate toxicity, *n* = 1; azathioprine toxicity, *n* = 1; cardiac liver, *n* = 1; congenital hepatic hemi-atrophy, *n* = 1; portal fibrosis, *n* = 1; and haemangioma associated with fibrosteatosis, *n* = 1. Of note, the latter two patients had positive serology for hepatitis C virus infection but without any viral replication found on repeated PCR; thus, active infection by hepatitis B and C virus was ruled out in all patients.

The median dry weight of the 15 deparaffinised histological blocks of transjugular biopsies was 1.13 mg (range: 0.23–2.06 mg), precluding any reliable chemical measurement of iron due to their small size [42].

Eighteen SIR MRI examinations were performed together with 18 R2* relaxometries (classical) close to the liver biopsy, whereas R2* relaxometry (IDEAL-IQ^®^) was available more recently and was carried out concurrently with liver histology in the last 12 cases.

MRI was performed with a median lag time of 30 days (range: 1–80) from the liver biopsy. The median LIC by SIR MRI was 60 µmol/g dry weight (range: 20–290); seven SIR MRIs were within the normal range, whereas 11 showed increased LIC > 40 µmol/g dry weight. The median LIC by R2* relaxometry (classical) was 14.35 ms (range: 4.7–35); nine R2* relaxometries (classical) were within the normal range, whereas nine R2* examinations showed increased LIC < 15 ms. The median LIC by R2* relaxometry (IDEAL-IQ^®^) was 12.30 ms (range: 1.8–28); six R2* relaxometry (IDEAL-IQ^®^) examinations were within the normal range, whereas six R2* IDEAL-IQ^®^s showed increased LIC < 15 ms.

### 3.2. Correlations between Quantitative MRI Methods and Histological Scoring Systems

LIC MRI with the three different methods was highly correlated with Deugnier’s and Turlin’s histological scoring (SIR: *r* = 0.8329, *p* = 0.0002; R2* relaxometry (classical): *r* = −0.9099, *p* < 0.0001; R2* relaxometry (IDEAL-IQ^®^): *r* = −0.872, *p* = 0.0018) (Table 3 and Figure 1, Figure 2 and Figure 3).

LIC determined with the three MRI methods was also significantly correlated with Scheuer’s classification (SIR: *r* = 0.7106, *p* = 0.0009; R2* relaxometry (classical): *r* = −0.7453, *p* = 0.0004; R2* relaxometry (IDEAL-IQ^®^): *r* = −0.7142, *p* = 0.0120) (Table 3).

Splenic iron load (determined by R2* relaxometry) was significantly correlated with both Deugnier’s and Turlin’s histological scoring (*r* = −0.6108, *p* = 0.0176) and to a lesser extent with Scheuer’s classification (*r* = −0.4761, *p* = 0.0458).

There was a high correlation between LIC measured by the three MRI methods (SIR and R2* relaxometry (classical): *r* = −0.8283, *p* < 0.0001; SIR and R2* relaxometry (IDEAL-IQ^®^): *r* = −0.9193, *p* < 0.0001; R2* relaxometry (classical) and R2* relaxometry (IDEAL-IQ^®^): *r* = 0.9231, *p* < 0.0001) (Figure 4, Figure 5 and Figure 6).

### 3.3. Evaluation of the Accuracy of the Three Quantitative MRI Methods for the Diagnosis of Iron Overload in Haemodialysis Patients

The sensitivity of the three MRI methods ranged from 62.5% (R2* relaxometry (IDEAL-IQ^®^)) to 80% (SIR) with an intermediate value for R2* relaxometry (classical) (70%), whereas the specificity was equal to 99% for these three MRI methods (Table 4).

The PPV for the diagnosis of iron overload was also high for the three MRI methods (99%), associated with high NPV, although these differed slightly between MRI methods: SIR (71.83%), R2* relaxometry (classical) (62.96%) and R2* relaxometry (IDEAL-IQ^®^) (57.63%) (Table 4).

The positive likelihood ratio (PLR) for the diagnosis of iron overload was very high for the three MRI methods, although slight differences were observed—PLR = 80 for SIR, PLR = 70 for R2* relaxometry (classical) and PLR = 62.5 for R2* relaxometry (IDEAL-IQ^®^)—and the posterior probability of having iron overload was also very high for these three MRI methods (99%) (Table 4). Fagan’s nomograms for these three MRI methods are presented in Figure 7.

## 4. Discussion

This study demonstrates excellent agreement between LIC values determined by three different MRI methods (SIR according to Rennes University, R2* relaxometry (classical) and R2* relaxometry with multi-peak spectral modelling) and quantitative histological estimation of iron load in 16 haemodialysis patients (studied for 18 liver biopsies). In this study, we used Deugnier’s and Turlin’s histological scoring with Perls’ staining, which has been validated in both haemochromatotic and non-haemochromatotic iron overload disorders and has been shown to accurately quantify the range of iron overload from mild forms, as in various non-haemochromatotic liver diseases, to severe forms, as in homozygous genetic haemochromatosis [30,31,32,33,34].

This study extends our previous results for the first 11 patients who were analysed only by SIR MRI and with Scheuer’s classification with Perls’ stain [43].

In this present study, we determined the diagnostic accuracy of the three MRI methods for detecting iron overload in dialysis patients and showed their high specificity (99%) and fairly good sensitivity (range: 62.5%–80%). Using the results of a recent pool analysis, including 10 quantitative MRI studies and one susceptometry study totalling 500 haemodialysis patients, showing a prevalence of haemodialysis-associated haemosiderosis of 66% [11], we determined the PPV of the three MRI methods as 99%, together with a NPV range of 57%–72% depending on the method [38]. All these statistical parameters suggest the high accuracy of these three MRI methods for the diagnosis of iron overload, which is strongly reinforced by the positive likelihood ratio, a powerful tool for summarising the diagnostic accuracy of a test that represents the direction and strength of evidence provided by a test result [38,39]. It is the ratio of the probability of the specific test result in patients who have the disease to the probability in those who do not have the disease. The values of the PLR range from zero to infinity. The farther the PLR is away from a value of one, the stronger the evidence provided by the test [38,39]. This later parameter ranged from 62.5 for R2* relaxometry (IDEAL-IQ^®^) to 80 for SIR MRI, and was intermediate (70) for R2* relaxometry (classical) [38,39]. Thus, considering that a PLR > 10 provides strong evidence for a diagnosis in most circumstances, the very high likelihood ratio found in this study strongly reinforces the accuracy of the three MRI methods for the evaluation of liver iron load in dialysis patients as evidenced in Fagan’s nomograms [40]. This study also respected the ethical requirements for frail haemodialysis patients in whom liver biopsy was only required for medical management and MRI was performed for the evaluation of their liver disease.

Our study has some limitations. The first relates to the small sample size analysed here—this issue is intrinsically linked to the context of ESKD associated with frailty and increased haemorrhagic risk, limiting liver biopsy to mandatory cases explaining the previous suggestion of Rostoker, Vaziri and Fishbane in 2016 [4] and more recently in 2018 by a panel of experts (from a conference devoted to positive iron balance in ESKD [44]) on the value of gathering a collection of liver biopsies from dialysis patients. This situation has some analogy with the anatomic demonstration of accuracy of R2* MRI for the diagnosis of ferric cardiomyopathy performed in a small panel of thalassaemic patients (*n* = 12) either during heart transplantation or at autopsy [45]. The second relates to the intrinsic limitations of MRI techniques for measurement of liver iron content. In addition to the importance of the skill of the radiologic team (both technician and radiologist) for the quality of the examinations, each technique has its own limitation: R2 relaxometry requires regular calibration of the apparatus by phantoms, is prone to respiratory motion artefacts and is influenced by steatosis; R2* is influenced both by iron and fat (explaining the development of multipeak spectral modelling methods) and its calibration is also controversial; and SIR requires a careful reproduction of Rennes University technical modalities and exhibits a loss of linearity at LIC values > 350 µmol/g (a situation rarely seen in haemodialysis-associated haemosiderosis) [15,19,46,47,48]. Lastly, the variability of hepatic concentration in some liver diseases can also influence MRI measurement [42].

Thus, considering the results of this study, especially the positive likelihood ratio and Fagan’s nomograms, we advocate first for its ease of use as a tool for the detection of iron overload in the setting of dialysis the use of R2* relaxometry; second, in positive cases, taking into consideration the actual high prevalence of non-alcoholic fatty liver disease in the dialysis population [49] as the ability of excessive iron therapy to induce liver steatosis in ESKD [23], it seems of paramount importance to quantify the liver fat fraction and to obtain a reliable evaluation of liver iron content in this setting by multi-peak spectral modelling; third, in the case of confirmed liver iron overload (without important fat fraction), a quantification of LIC by SIR MRI is of high diagnostic interest; fourth, in case of severe iron overload on SIR MRI (LIC > 200 µmol/g of dry liver), a search for pancreatic and heart iron deposits is mandatory by R2* relaxometry.

In summary, quantitative MRI methods (SIR MRI according to Rennes University), R2* relaxometry (classical) and R2* relaxometry (IDEAL-IQ^®^) with multi-peak spectral modelling can reliably diagnose iron overload in haemodialysis patients. Further, this study contributes to overcoming the knowledge gap identified by the KDIGO conference regarding validation of MRI techniques for quantifying LIC in ESKD patients [25].

## 5. Conclusions

Quantitative MRI techniques can be used as routine tools to quantify liver iron content in haemodialysis patients and for monitoring iron stores while on parenteral iron therapy.

## Figures and Tables

**Figure 1 jcm-09-00017-f001:**
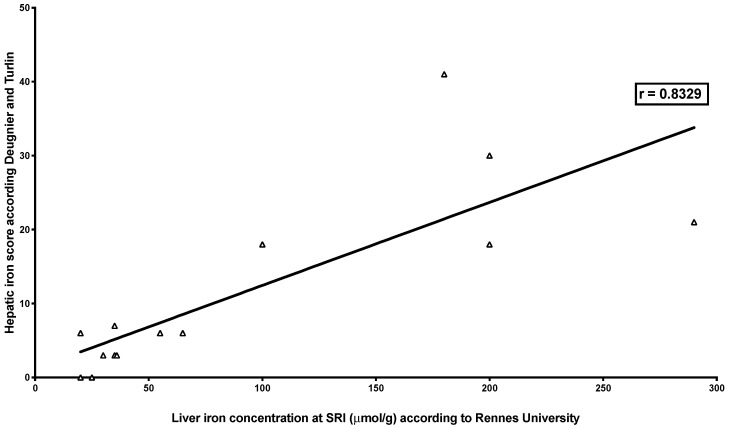
Correlation between liver iron concentration measured by SIR MRI and Deugnier’s and Turlin’s histological scoring. The figure shows the relationship between LIC measured by SIR MRI and iron histological scoring successfully performed in 15 liver biopsies (out of 18). The relationship was studied by Spearman’s rank correlation test, which showed a very high correlation (*r* = 0.8329; *p* = 0.0002).

**Figure 2 jcm-09-00017-f002:**
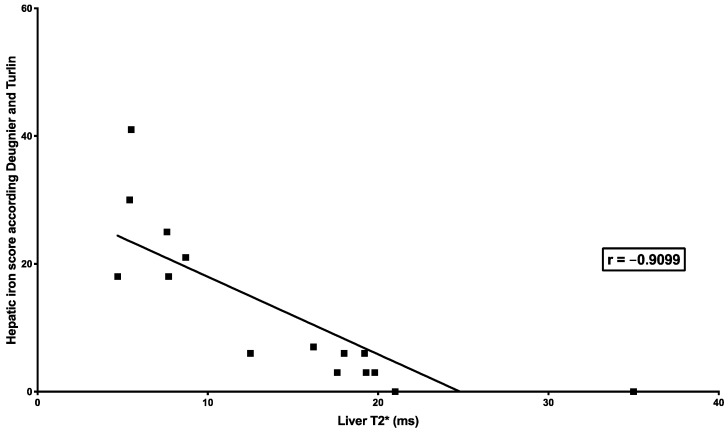
Correlation between liver iron concentration measured by R2* relaxometry (classical) MRI and Deugnier’s and Turlin’s histological scoring. The figure shows the relationship between T2* measured by R2* relaxometry (classical) and iron histological scoring successfully performed in 15 liver biopsies (out of 18). The relationship was studied by Spearman’s rank correlation test, which showed a very high correlation (*r* = −0.9099; *p* < 0.0001).

**Figure 3 jcm-09-00017-f003:**
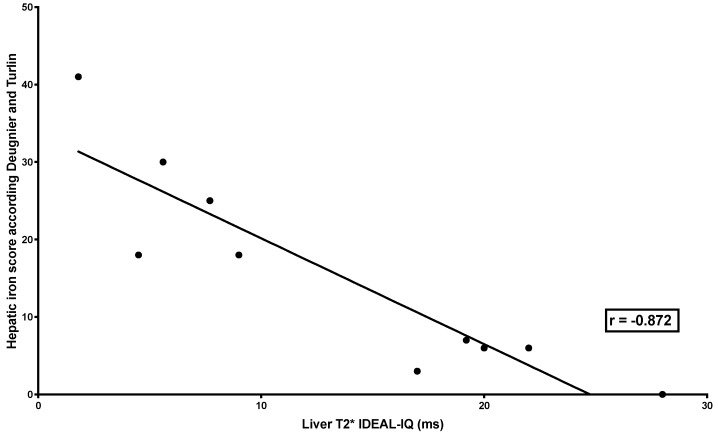
Correlation between liver iron concentration measured by R2* relaxometry (IDEAL-IQ^®^) MRI and Deugnier’s and Turlin’s histological scoring. The figure shows the relationship between T2* measured by R2* relaxometry (IDEAL-IQ^®^) and iron histological scoring successfully performed in 10 liver biopsies (out of 12). The relationship was studied by Spearman’s rank correlation test, which showed a very high correlation (*r* = −0.872; *p* = 0.0018).

**Figure 4 jcm-09-00017-f004:**
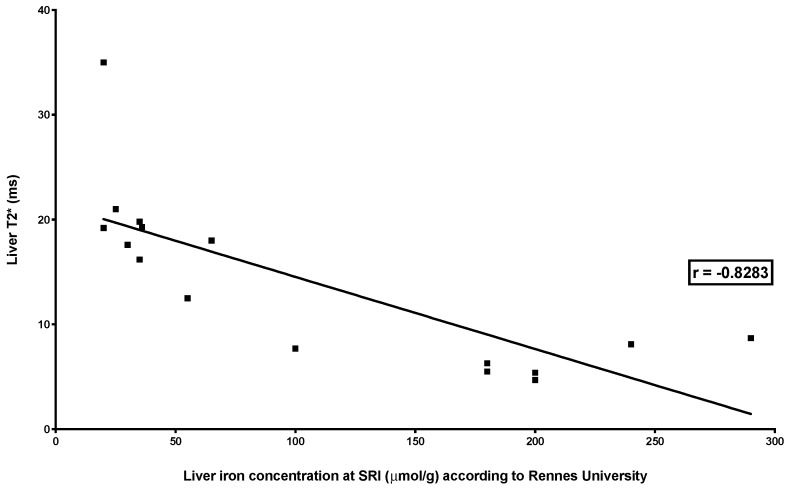
Correlation between liver iron concentration measured by SIR MRI and R2* relaxometry (classical). The figure shows the relationship between LIC measured by SIR and T2* measured by R2* relaxometry in 18 combined MRI exams. The relationship was studied by Spearman’s rank correlation test, which showed a very high correlation (*r* = −0.8283; *p* < 0.0001).

**Figure 5 jcm-09-00017-f005:**
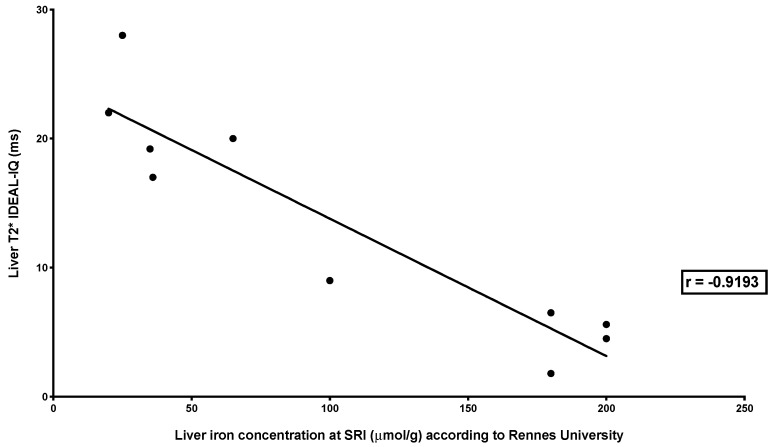
Correlation between liver iron concentration measured by SIR MRI and R2* relaxometry (IDEAL-IQ^®^). The figure shows the relationship between LIC measured by SIR and T2* measured by R2* relaxometry (IDEAL-IQ^®^) in 12 combined MRI exams. The relationship was studied by Spearman’s rank correlation test, which showed a very high correlation (*r* = −0.9193; *p* < 0.0001).

**Figure 6 jcm-09-00017-f006:**
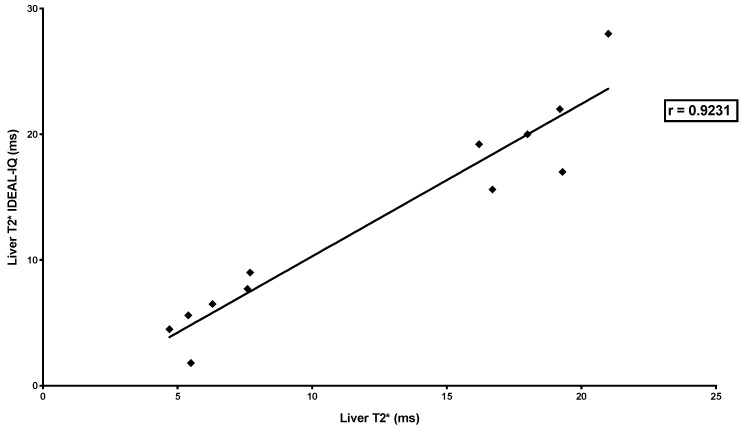
Correlation between liver iron concentration measured by R2* relaxometry (classical) and R2* relaxometry (IDEAL-IQ^®^). The figure shows the relationship between T2* measured by R2* relaxometry (classical) and T2* measured by R2* relaxometry (IDEAL-IQ^®^) in 12 combined MRI exams. The relationship was studied by Spearman’s rank correlation test, which showed a very high correlation (*r* = 0.9231; *p* < 0.0001).

**Figure 7 jcm-09-00017-f007:**
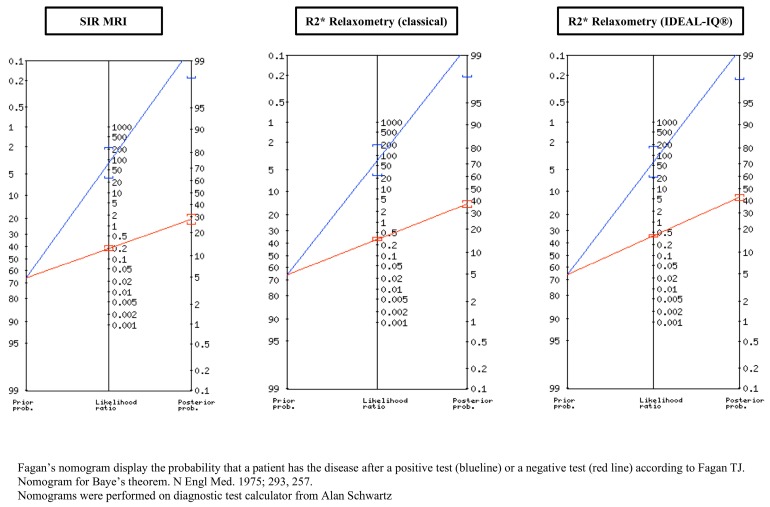
Fagan’s nomograms of SIR MRI, R2* relaxometry (classical) and R2* relaxometry (IDEAL-IQ^®^) for the diagnosis of iron overload in haemodialysis patients.

**Table 1 jcm-09-00017-t001:** Liver iron storage according to Deugnier’s and Turlin’s histological scoring [26].

Site	Scoring	Final Score
Hepatocytic iron score (HIS)	0, 3, 6, 9, or 12 according to granule size in each Rappaport area	0–36
Sinusoidal iron score (SIS)	0, 3, 6, 9, or 12 according to granule size in each Rappaport area	0–12
Portal iron score (PIS)	0, 1, 2, 3, or 4 according to the percentage of iron overloaded macrophages, biliary cells and vascular walls	0–12
Total iron score (TIS)		0–60

**Table 2 jcm-09-00017-t002:** Hepatocytic liver iron storage according to Scheuer’s classification [27].

Grade	Histology
Grade 0	Granules absent or barely discernible at ×400 magnification
Grade 1	Granules barely discernible at ×250 magnification and easily confirmed at ×100
Grade 2	Discrete granules resolved at ×100 magnification
Grade 3	Discrete granules resolved at ×25 magnification
Grade 4	Masses visible at ×10 magnification or by naked eye

**Table 3 jcm-09-00017-t003:** Correlation between the three quantitative magnetic resonance imaging (MRI) methods and two histological scoring systems for liver iron storage (Spearman’s rank correlation test).

Histological Scoring of LIC	SIR According to Rennes University	R2* Relaxometry (Classical)	R2* Relaxometry (IDEAL-IQ^®^)
Deugnier’s and Turlin’s histological scoring	*r* = 0.8329 *p* = 0.0002	*r* = −0.9099 *p* < 0.0001	*r* = −0.872 *p* = 0.0018
Scheuer’s classification	*r* = 0.7106 *p* = 0.0009	*r* = −0.7453 *p* = 0.0004	*r* = −0.7142 *p* = 0.0120

LIC, liver iron concentration; SIR, signal intensity ratio; IDEAL-IQ^®^, Iterative Decomposition of water and fat with Echo Asymmetry and Least-squares estimation.

**Table 4 jcm-09-00017-t004:** Diagnostic accuracy of the three MRI methods for haemodialysis-associated haemosiderosis.

	SIR MRI According to Rennes University	R2* Relaxometry (Classical)	R2* Relaxometry (IDEAL-IQ^®^)
Sensitivity (%)	80	70	62.5
Specificity (%)	99	99	99
Positive predictive value (%)	99.36	99.27	99.18
Negative predictive value (%)	71.83	62.96	57.63
Positive likelihood ratio	80	70	62.5
Posterior probability (%)	99.36	99.27	99.18

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
