# Peer review of "Histological Scores Validate the Accuracy of Hepatic Iron Load Measured by Signal Intensity Ratio and R2* Relaxometry MRI in Dialysis Patients"

_jcm, 2019, doi:10.3390/jcm9010017_

Round 1
Reviewer 1 Report
The authors investigated the performance of 3 different MRI based techniques to quantitate the liver iron in patients with chronic renal failure (CRF) underwent chronic HD.
The methods includes, 1) signal intensity ratio (SIR) in group 1 , 2) classical R2* in group 2, and 3) Ideal IQ in group 3. The values were correlated with the Liver iron values measured from histologic specimens.
Group 1 and 2 include in 18 pts and group 3 12 pts. In group 1, 7 were normal, 11 were abnormal per histopathologic values, in group 2, 9 normal and 9 abnormal, in group 3, 6 were normal and 6 were abnormal. The liver iron values from all 3 MRI techniques area were correlated well with the iron values measured from histopathology. The sensitivity and septicity of the 3 MRI techniques were ranged 62.5 - 80 % and 99%, respectively.
The MRI based liver iron management techniques have been used in patients with hematologic malignancies, such as leukemia, other acquired and congenital iron deposition disease. The absolute value of liver iron is known to inversely correlated with the transverse relaxation times, T2*, derived from the R2* curves. The results of this study is not unexpected.
The work is done in an organized fashion and the manuscript flows well. Comments and recommendations are followings.
The liver iron measurement is highly depending on MRI acquisition technique. The MRI techniques, including the MRI imaging parameters, are not included in the method section. This informatization will also important to reader to reproduce the results. Sample size is too small to lead to a meaningful conclusions. Discuss limitations of the techniques in measuring the accurate liver iron values.
C. Wood, C. Enriquez, N. Ghugre et al., “MRI R2 and R2∗ mapping accurately estimates hepatic iron concentration in transfusion-dependent thalassemia and sickle cell disease patients,” Blood, vol. 106, no. 4, pp. 1460–1465, 2005.
Hou P1, Popat UR, Lindsay RJ, Jackson EF, Choi H. A practical approach for a wide range of liver iron quantitation using a magnetic resonance imaging technique. Radiol Res Pract. 2012.
Reviewer 2 Report
The study investigate the possible use of quantitative MRI methods non-invasive diagnosis of iron overload in haemodialysis patients. The description of the methodology appears clear and structured. The level of detail is sufficient and the study is clinically relevant and deals with on open challenge in the field. I would recommend to publish the paper after the following comments are addressed.
Comments:
The outcome of the paper should be sharper. Stating that the “This study supports the accuracy []” is to the Reviewer not clear. Would you consider to change this accordingly to your findings? Maybe: “This study recommend the use of XYZ for XYZ as it can give XYZ accuracy/precision/etc”. What could be the effect of using only one radiologist to perform the scans? Would this study be under any circumstance be considered as operator dependent?
Reviewer 3 Report
The manuscript of ROSTOKER et al explores correlations between quantitative MRI iron measurements and histological methods in dialysis patients. The manuscript is well written. Clarification of questions raised below might help to improve this paper.
Abstract (lines 25-26): The phrase “16 haemodialysis patients in whom a liver biopsy (n=18)” is confusing. Why number of patients and number of biopsy do not match? Methods: It is unclear how long the studied patients were on iron therapy before the MRI exam and liver biopsy. Part “2.2. Quantification of hepatic iron stores by MRI”: There is a repetitive phrase in several paragraphs: “… same MRI machine (GE Medical Systems) operating at a field strength of 1.5 Tesla according to the manufacturer’s recommendations…” The scanner description can be provided once at the beginning of this part. The last paragraph in the part “2.2. Quantification of hepatic iron stores by MRI” (lines 132-136) is irrelevant to the scientific problem discussed in the manuscript. It is uncommon to have a financial explanation in the research manuscript. Part 3 (Results). Correlation coefficients “rho” are usually presented in the papers as r. Figures 1-6: statistical method is usually presented in the figure’s legend; only r value is usually shown on the graph itself. Figures 1-6: it is unclear why number of measurements are different in different figures. Any reason for exclusions? Figure 9 repeats data shown on the Table 4. The bar-graph on the figure 9 does not provide any additional information to the data on the Table 4, therefore, it should be removed. Results shown in the Figure 8 need to be explained in better details. The discussion part is too brief. It might be improved by comparison of the current findings with the published data, discussion of next steps and perspectives. The last sentence of the Discussion (lines 310-315) is 6 lines long and very confusing. Get to the point! Can any suggestion be made regarding which MRI method might be preferable in measuring liver iron, based on the study results?
Round 2
Reviewer 1 Report
The Authors have addressed my comments, but I still feel the claim of the paper should be sharper - what may be clear for you can be not clear for a broader.
Example of this, as already pointed out in my previous revision are the statement of the research question in the last paragraph of the introduction, the last paragraph of the discussion and the conclusion.
Stating: "the results of this study confirm the accuracy of quantitative MRI methods" is very vague and does not allow one to take the message home.
Again, stating "Our results support the accuracy of quantitative MRI methods" is not very clear why and how.
I would suggest you to ask a non expert to read this and to have his comment as well.
Author Response
the last comments of the reviewer 1.
We have sharply rewritten the last paragraph of the introduction, the last paragraph of the discussion and the conclusion as advised, to make easily readable and understandable our article by the broad readership of the Journal of clinical Medicine. In this second revised version, we have also included English language modifications advised by the translation division of MDPI.
We hope that this last revised version will fulfill the high standard of publication of the Journal of Clinical Medicine.
Reviewer 2 Report
The manuscript of Rostoker et al has improved significantly in the revised version. All suggestions from the reviewers were implemented. More details were added to methods. Discussion part is expanded.
Author Response
Thank you a lot for your review comments.
This manuscript is a resubmission of an earlier submission. The following is a list of the peer review reports and author responses from that submission.